# GP IIb/IIIa Receptor Inhibitors in Mechanically Ventilated Patients with Cardiogenic Shock due to Myocardial Infarction in the Era of Potent P2Y12 Receptor Antagonists

**DOI:** 10.3390/jcm11247426

**Published:** 2022-12-14

**Authors:** Vojko Kanic, Gregor Kompara, David Suran

**Affiliations:** Department of Cardiology and Angiology, Division of Internal Medicine, University Medical Center Maribor, 2000 Maribor, Slovenia

**Keywords:** cardiogenic shock, GP IIb/IIIa receptor inhibitor, mechanical ventilation, myocardial infarction, P2Y12 receptor antagonist, outcome

## Abstract

**Objective:** To investigate the association between GP IIb/IIIa receptor inhibitors (GPI) and mortality and bleeding in patients with cardiogenic shock (CS) due to myocardial infarction (MI) who were mechanically ventilated on admission. **Methods**: We retrospectively divided 153 patients into two groups (with or without GPI). Thirty-day and one-year all-cause mortality and bleeding were studied. **Results**: The observed 30-day and one-year all-cause mortality were similar in both groups [54 (69.2%) with GPI vs. 62 (82.7%) without GPI; *p* = 0.06, and 60 (76.9%) with GPI vs. 64 (85.3%) without GPI; *p* = 0.22, respectively]. Patients with GPI suffered fewer unsuccessful PCI (TIMI 0/1 was 10% in the GPI group vs. 57% in the group without GPI), experienced more improvements in TIMI ≥ 1 flow [68 (87.2%) in the GPI group vs. 38 (50.7%) without GPI; *p* < 0.0001], and they achieved better cerebral performance category (CPC) scores (1.61 ± 0.99 with GPI vs. 2.76 ± 1.64 without GPI; *p* = 0.005). The bleeding rate was similar in patients with and without GPI [33 (42.3%) vs. 31 (41.3%): *p* = 1.00], in patients with P2Y12 receptor antagonists (P2Y12) [18 (46.1%) with GPI vs. 21 (46.7%) without GPI; *p* = 1.00], and in patients with potent P2Y12 [8 (30.8%) with GPI vs. 9 (37.5%) without GPI; *p* = 0.77]. **Conclusions**: Due to the study design (limited sample size, retrospective inclusion with high risk of selection bias), our analysis does not allow us to draw conclusions about the effectiveness of GPI in this context. Despite all these limitations, GPI were associated with improved TIMI flow after PCI in our multivariable model without increasing bleeding rates. In addition, better CPC scores were observed, but no association between GPI and outcome was found. Our analysis suggests that selective use of GPI may be beneficial in mechanically ventilated patients with MI in CS without additional bleeding risk, even in the era of potent P2Y12.

## 1. Introduction

Antithrombotic treatment and reperfusion are essential for patients with myocardial infarction (MI) and cardiogenic shock (CS) [1,2,3,4,5,6,7,8,9]. Despite optimal treatment, the mortality rate in these patients is still very high [4,10,11]. The most effective antithrombotic treatment with the lowest bleeding risk, especially in patients with CS, is still unknown [6,10,12,13]. Glycoprotein IIb/IIIa inhibitors (GPI) are the most effective antiplatelet agents [9,13,14], although they are currently only recommended for periprocedural complications in patients with myocardial infarction (MI) when there is evidence of a thrombotic complication or no reflow [1,2,5,12]. Their use has been associated with higher bleeding rates [1,2,3,12,14]. In the era of potent oral P2Y12 receptor antagonists (P2Y12) (prasugrel, ticagrelor), the routine adjunctive use of GPI is not recommended [1,2,15]. However, this recommendation is not supported by any randomized trial [5]. In certain groups of patients (unconscious and/or intubated patients, patients unable to swallow tablets), oral administration of the drugs is delayed or sometimes even impossible [4,10]. Even if oral administration is feasible, the antiplatelet effect of orally administered drugs is delayed by gastroparesis, delayed intestinal absorption, and slower metabolism of the drugs due to compromised hemodynamics in cardiogenic shock, hypothermia, and after morphine administration [6,7,13,16,17,18]. This may be overcome with aspirin, which can be administered intravenously, and another potent and rapidly acting intravenous antiplatelet agent that can act immediately [3,9,13,17,19]. Parenteral antiplatelet agents include cangrelor (a P2Y12) and GPI [4,6,7]. There are data indicating that cangrelor is associated with a similar mortality outcome to oral P2Y12 in patients with CS [4]. No randomized data are available for cangrelor in combination with potent oral P2Y12 [15]. Its high cost also hinders routine application [3]. On the other hand, a randomized registry of patients with ST-elevation MI [5], a meta-analysis in patients with CS, and some recent retrospective data have shown that GPI could be an effective and beneficial adjunctive therapy for such patients [7,9,13], although most of the data are from the pre-prasugrel/ticagrelor era [3,7,9,13]. According to these data, we assume that patients who cannot take P2Y12 orally may benefit from treatment with GPI. This would include patients with CS who are already mechanically ventilated on hospital admission [4]. However, there are no patient data on the adjunctive use of GPI in these patients. Therefore, we aimed to investigate whether the adjunctive use of GPI is associated with 30-day and one-year outcomes and bleeding in patients with CS who were already mechanically ventilated on admission in the era of potent P2Y12.

## 2. Materials and Methods

We retrospectively examined the hospital database of the University Medical Center Maribor, a tertiary referral hospital with a 24/7 PCI service, from January 2007 to December 2018. Of the 8348 consecutively examined MI patients, we found 649 patients who either suffered from CS or were mechanically ventilated. We excluded 496 patients who died before PCI, suffered from CS only, were ventilated only, developed CS after hospital admission, were mechanically ventilated during hospitalization, or were treated conservatively. The final study cohort included 153 patients who suffered from CS and were already mechanically ventilated on admission or were mechanically ventilated immediately upon admission but before PCI (Figure 1). Patients were divided into two groups—one group that received GPI (78 patients) and one group that did not (75 patients). These groups were compared, and all-cause mortality was assessed at 30 days and one year.

### 2.1. Patients and Definitions

The diagnosis of MI was made in accordance with published guidelines, which were also followed in the management of patients [1,2].

The diagnosis of CS was made based on the accepted definition of a systolic blood pressure ≤90 mm Hg for ≥30 min or the need for supportive measures to maintain a systolic blood pressure of >90 mm Hg, clinical signs of pulmonary congestion, and signs of end-organ hypoperfusion.

Cardiac arrest was defined as cardiopulmonary collapse with associated loss of pulse requiring chest compressions or rescue shock by a health professional. We did not exclude patients based on their initial cardiac rhythm or the duration of advanced cardiac life support. Only cardiac arrests outside of the hospital or immediately after admission but before PCI were used as a covariate in regression analysis. The decision to provide mechanical ventilation was at the discretion of the pre-hospital emergency physician or the physician in the admitting department. The intra-aortic balloon pump (IABP) was mostly used for mechanical circulatory support. Extracorporeal membrane oxygenation (ECMO) was used in only four (2.6%) patients (two patients in each group), all of whom died. Thrombolysis in myocardial infarction (TIMI) flow grades were used to assess coronary blood flow [20]. Bleeding events were classified using the Bleeding Academic Research Consortium (BARC) definition and BARC 3–5 bleeding was documented [21].

Mild induced therapeutic hypothermia in patients with out-of-hospital cardiac arrest was recommended. Their temperature was maintained between 32 °C and 34 °C for 24 h. The cerebral performance category (CPC) score was assessed as described previously [22].

Troponin levels were determined at admission and at least once in the first 24 h. After that, serum troponin levels were measured at various times at the discretion of the treating physician. Troponin I was determined by the chemiluminescence immunoassay method on Siemens Dimensions Vista Systems (Siemens Healthcare Diagnostics, DE, USA) with the reference interval < 0.045 µg/L. The highest serum troponin level during the entire hospitalization was defined as the peak troponin level. The ventricular ejection fraction was assessed by bedside echocardiography in the first 48 h after admission.

### 2.2. Pharmacological Treatment

In patients without contraindications, 500 mg of aspirin orally or, if this was not possible, 300 mg intravenously and enoxaparin 1 mg/kg intravenously were administered at first medical contact until 2011. After 2011, unfractionated heparin 5000 IU was usually administered intravenously. P2Y12 was administered upstream in ST-elevation MI, if possible, at the discretion of the emergency physician. Some patients deteriorated and were intubated during transport after receiving treatment. A loading dose of clopidogrel 300–600 mg was used until 2011, after which prasugrel 60 mg or ticagrelor 180 mg was usually used. GPI were not used upstream. The treatment of patients who came directly to our center was basically the same as in the pre-hospital setting. In these patients, bivalirudin (with a bolus of 0.75 mg/kg and an infusion of 1.75 mg/kg/h), or heparin or enoxaparin, was administered at the discretion of the admitting physician. All patients received aspirin (orally or intravenously). Otherwise, treatment with other drugs was the same as in the pre-hospital setting. Administration of the more potent P2Y12 in addition to the clopidogrel loading dose before hospitalization was not common. The strategy of angioplasty, PCI of other coronary arteries, or the adjunctive use of GPI (eptifibatide or abciximab) was at the discretion of the operator, and we did not distinguish between the planned and bail-out use of GPI. Aspirin was recommended to be continued indefinitely, and P2Y12 were mostly recommended for 12 months. However, patients who required additional anticoagulation or who were at high risk of bleeding were treated individually (depending on the year of admission and the bleeding risk). The data on GPI, aspirin, and P2Y12 were provided for all patients, and data on all other essential patient and procedural characteristics were at least 96% complete. Ascertainment of endpoints was 100% complete. The data on death were provided by the Slovenian National Cause of Death Registry. Ethical governance, and waiver of consent approvals were granted by the University Medical Center Maribor Committee for Medical Ethics (Reference: UKC-MB-KME-59/19), and all methods were performed in accordance with the requirements of the Declaration of Helsinki.

### 2.3. Study End Points

The end points of the study were all-cause mortality at day 30 and one year. We also compared the in-hospital bleeding rate in both groups and the bleeding rate in patients who received GPI and P2Y12. In addition, we assessed the CPC score at the end of hospitalization.

### 2.4. Statistical Methods

The distributions of continuous variables in the two groups were compared with either the two-sample *t*-test or the Mann–Whitney test, depending on whether the data followed the normal distribution. The distributions of the categorical variables were compared with the Chi-square test. We counted the end point events that occurred during the follow-up period and compared their rates between cohorts of patients who did or did not receive GPI. Follow-up began on the day of admission and lasted until the day of death or up to one year, whichever occurred first. We generated Kaplan–Meier curves for patients with and without GPI for unadjusted mortality. Binary logistic regression models were performed using the enter mode to identify associations between GPI and 30-day mortality and bleeding, and Cox proportional hazards regression was used to calculate hazard ratios (HRs) as estimates of one-year mortality. In the regression analyses of mortality, we controlled for age, sex, glomerular filtration rate, diabetes, hypertension, hyperlipidemia, ST-elevation MI, cardiopulmonary resuscitation, hypothermia, P2Y12, TIMI 0/1 flow after PCI, bleeding, and GPI. Model covariates were predefined within the study design based on their clinical and pathophysiologic relevance as a confounder, significance in the literature, and frequency of occurrence in this cohort of patients.

In the analyses of bleeding, we performed two multivariable models, one with the P2Y12 variable (clopidogrel, prasugrel, ticagrelor) and one with the potent P2Y12 variable (prasugrel, ticagrelor), to separately assess the influence of GPI on bleeding in patients with potent P2Y12. Data were analyzed using SPSS 25.0 software for Windows (IBM Corp., Armonk, NY, USA). All *p*-values were two-sided, and values less than 0.05 were considered statistically significant.

## 3. Results

Of the 78 patients who received GPI, 7 (8.9%) received abciximab, and the remainder, eptifibatide. Patients receiving GPI were more likely to have undergone therapeutic hypothermia, have more extensive coronary artery disease (requiring multivessel PCI), PCI of LAD, and have achieved better TIMI flow after PCI. Patients with GPI were also more likely to have experienced improvement in TIMI grade flow ≥ 1 during PCI. They tended to have had fewer previous strokes. Their GFR was more likely to be higher. Otherwise, the groups were similar in terms of age, risk factors, ST-elevation MI, cardiopulmonary resuscitation, use of mechanical circulatory support, and frequency of use of P2Y12. Potent P2Y12 were used in 60% (50 out of 84 patients) of patients who received P2Y12. Baseline data and procedural characteristics of the study population are shown in Table 1. The percentage of concomitant GPI use differed by year of treatment. However, the year of admission was not associated with either 30-day (*p* = 0.066) or one-year mortality (*p* = 0.18).

### 3.1. Thirty-Day Mortality

After 30 days, 116 (74.8%) patients had died. The observed 30-day all-cause mortality was similar in both groups [54 (69.2%) patients with GPI died vs. 62 (82.7%) without GPI; *p* = 0.06]. After adjustment for confounders, GPI were not associated with 30-day mortality risk (OR 0.52; 95%CI 0.19–1.46; *p* = 0.22). Bleeding and P2Y12 were associated with 30-day mortality (Table 2). The Hosmer-Lemeshow goodness-of-fit test showed that the model was a good fit to the data (Chi-square = 6.62, df = 8, *p* = 0.578). An analysis of the collinearity of the variables yielded a variance inflation factor (VIF) of all included variables of <1.7.

### 3.2. Mortality after One Year

After one year, 124 (81.0%) patients had died, with deaths being similarly distributed between the two groups [60 (76.9%) patients with GPI died compared to 64 (85.3%) patients without GPI; *p* = 0.22]. The constructed Kaplan–Meier curve also showed similar unadjusted survival curves (log Rank = 0.082) (Figure 2). After adjustment for confounders, the association between GPI and one-year mortality risk was nonsignificant (adjusted HR 0.92; 95%CI 0.62 to 1.38; *p* = 0.70). Only P2Y12 were associated with one-year mortality risk (Table 2).

### 3.3. TIMI Grade Flow

We checked the TIMI grade flow before and after PCI and the improvement of TIMI grade flow ≥ 1 after PCI (Table 3). The TIMI grade flow before PCI was similar in all groups. GPI resulted in significantly better TIMI grade flow after PCI in all groups, regardless of whether they received P2Y12 or even potent P2Y12. In addition, improvement of TIMI grade flow ≥ 1 was more frequent in all groups receiving GPI, regardless of whether they received P2Y12 or potent P2Y12.

### 3.4. Bleeding

BARC 3–5 in-hospital bleeding occurred in 64 (41.8%) patients. The rate of bleeding was similar in patients receiving GPI and those without GPI [33 (42.3%) patients with GPI compared to 31 (41.3%) patients without GPI; *p* = 1.00] (Table 1). Two BARC 4 bleeds were observed in the group without GPI. All other bleeds were BARC 3. We also examined the effects of adding GPI on bleeding risk in patients receiving an oral P2Y12 (clopidogrel, prasugrel, or ticagrelor). Of the 84 patients receiving a P2Y12, 45 patients (57.7% of patients receiving a P2Y12) also received GPI. Bleeding occurred with similar frequency in both groups [21 (46.7%) patients with GPI compared to 18 (46.2%) patients without GPI; *p* = 1.00] (Table 1). We also investigated whether the addition of GPI increased the bleeding rate in patients receiving potent P2Y12. Of the 50 patients who received potent P2Y12, 26 (52.0%) also received GPI. Bleeding rates were similar in patients with or without GPI [8 (30.8%) bleeds in patients receiving GPI vs. 9 (37.5%) in patients not receiving GPI; *p* = 0.76] (Table 1).

We created two models to evaluate the association between GPI and P2Y12 and bleeding. In model 1, all P2Y12 (clopidogrel, prasugrel, and ticagrelor) were used. After adjustment for confounders, neither P2Y12 nor GPl were independently associated with bleeding in patients (OR 0.86, 95% CI 0.40 to 1.84, *p* = 0.70; and OR 0.64, 95% CI 0.29 to 1.41, *p* = 0.27, respectively). Resuscitation, therapeutic hypothermia, and diabetes were associated with bleeding.

When we included only potent P2Y12 (prasugrel and ticagrelor) in the model, potent P2Y12, but not GPI, were associated with bleeding after adjustment (OR 2.57, 95% CI 1.10 to 5.99, *p* = 0.029; and OR 0.62, 95% CI 0.27 to 1.38, *p* = 0.24, respectively). In addition to potent P2Y12, cardiopulmonary resuscitation (OR 4.77; 95% CI 1.69 to 13.50; *p* = 0.003), therapeutic hypothermia (OR 5.52; 95% CI 2.04 to 14.92; *p* = 0.001) and diabetes (OR 0.35; 95% CI 0.12 to 0.99; *p* = 0.05) were associated with bleeding.

### 3.5. Cerebral Performance Category (CPC) Score

All included patients were in CS and were mechanically ventilated on admission. More than 70% (108) of patients underwent cardiopulmonary resuscitation before hospitalization or immediately after admission but before PCI, and 58 (53.7%) received GPI. Patients with GPI had better CPC scores than patients without GPI (1.57 ± 0.98 with GPI compared to 2.86 ± 1.44 without GPI; *p* = 0.011) (Table 1).

We did not assess the CPC score in resuscitated patients undergoing therapeutic hypothermia with respect to GPI because the number of survivors was small (21 (75%) patients with therapeutic hypothermia).

Interestingly, when we included the CPC score in the Cox regression model with the above variables, only the CPC score was associated with one-year survival (HR 1.67; 95% CI 1.36–2.06; *p* < 0.0001).

### 3.6. Complications during Hospitalization

Complications were similarly distributed in the groups. They are listed in Appendix A. As this was a retrospective study, we collected the diagnoses of complications from the electronic data as described by the treating physicians.

## 4. Discussion

There are no data on the potential effects of GPI in patients with MI and CS who were already mechanically ventilated on admission. These patients are among the most vulnerable and critically ill patients with extremely high mortality rates. We examined the potential impact of GPI on the outcomes in these patients, even in the era of potent P2Y12. The main findings of our analysis are:(a)Patients receiving GPI had significantly better TIMI flow after PCI and more often experienced an improvement of ≥ 1 TIMI grade flow during PCI.(b)GPI use was not associated with a higher bleeding rate in these patients.(c)The CPC score was better in patients with GPI.(d)Patients receiving GPI had similar 30-day and one-year mortality, and GPI were not independently associated with either outcome.

Our results suggest that the use of GPI improves TIMI flow after revascularization without increasing the risk of bleeding in these patients. As in previous observations, patients had a very high thrombotic burden evidenced by the high incidence of STEMI (88%) and TIMI flow 0/1 on admission (72.5%) [6]. Resuscitation is often performed in these patients (70% in our analysis), resulting in inflammation (which encourages platelet aggregation) and lactic acidosis (which reduces platelet inhibition) [6]. In addition, therapeutic hypothermia (37% of our patients) promotes platelet aggregation and stent thrombosis by weakening platelet inhibition [6]. Even in patients with ST-elevation MI without CS, effective platelet inhibition with oral drugs is mostly achieved within four hours [23,24]. In CS with decreased splanchnic perfusion, slowed intestinal motility, poorer intestinal absorption, and slower and poorer metabolism of drugs, effective platelet inhibition with oral drugs is expected to be even more delayed, and intravenous drugs are the only reliable choice to prevent peri-PCI thrombotic events [3,4,6,7,9,10,16,17]. Ideally, this potent, fast-acting platelet inhibition should not increase bleeding compared to oral antiplatelet drugs [4].

Previous data on patients with CS (some of whom were mechanically ventilated, although it is not clear whether mechanical ventilation began before/at or after PCI or later during hospitalization) are inconsistent and contradictory [7,8,9,13,25,26]. Our result is consistent with the previous finding of De Felice et al., where abciximab was not independently associated with one-year mortality in patients with CS [26]. However, only around 40% of patients were intubated in their study, and only clopidogrel was used [26]. In contrast to our result, some studies have found a clear association between GPI and one-year mortality [7,8]. Pooled data in a meta-analysis (with about 50% of ventilated patients) showed an association of GPI with lower 30-day and one-year mortality in CS [13]. The only prospective study in patients with CS showed no benefit of the routine use of abciximab compared to the selective use of abciximab (at the operator’s discretion) [25]. This was not a study of GPI vs. no-GPI but of the routine vs. selective use of GPI. However, none of these data included hypothermia and potent P2Y12 in the analysis. In addition, we used eptifibatide in 90% of patients.

Our result is also only partially consistent with our previous analysis, which, however, included different patient subsets (patients after out-of-hospital arrest and/or patients with CS) [9]. We found that GPI were associated with 30-day and one-year mortality in those patients. There were several differences between our observations. The data on the localization of the infarct, GFR, access site, diabetes, hypertension, hyperlipidemia, and CPC score were not available for our previous analysis, and the definition of bleeding was different. All of these parameters are known to influence the outcome. In addition, the patient selection was different. Comparisons must be made in light of these differences, and, in our opinion, these differences could explain our result.

Despite existing evidence that oral medications require more time to become bioavailable in these patients, the use of modern, more effective P2Y12 has been shown to be associated even with the short-term outcome [4,6,7,13,16,17,18]. However, previous reports at CS (except for our earlier observation [16]) have used virtually no potent P2Y12. Nevertheless, we observed more improvement in TIMI flow in patients with GPI after PCI (Table 1), indirectly confirming effective early platelet inhibition by parenterally administered drugs. In addition, there was a consistent improvement of TIMI grade of flow ≥ 1 after PCI, regardless of whether P2Y12 or even potent P2Y12 were used (Table 3). Better flow after PCI in patients with CS was also noted in some [7,9,13], but not all [8], previous results, and some meta-analyses in STEMI patients [3].

Patients receiving GPI had a higher thrombotic burden (more extensive coronary artery disease with more multivessel PCI and more therapeutic hypothermia). These patients would have been expected to suffer greater myocardial injury, and consequently lower ejection fraction after PCI. However, peak troponin levels and ejection fractions were similar in both groups (Table 1). We can assume that this was the result of more successful PCIs with better TIMI flow after intervention because of rapid and effective platelet inhibition in patients with GPI. It has been previously shown that better TIMI flow after PCI leads to better survival in patients with CS [27]. We surmise that the small sample size did not permit such a result. In our analysis, 30-day mortality also tended to be better in the group with GPI (*p* = 0.06).

Despite the higher thrombotic burden in the GPI group, the frequency of acute stent thromboses was similar in both groups (although fewer in number in the GPI group) (Table 1). All patients with stent thrombosis died. Early stent thrombosis may be related to residual target lesion thrombus, dissection, slow flow, under-expansion of the stent, or a combination of these factors, further emphasizing the importance of good platelet inhibition and better flow in these patients [28].

In addition, the CPC score was significantly better in the GPI patients. Although the number of patients was small, this result points in the right direction. Indeed, in these critically ill patients, the quality of survival is even more important than survival itself. However, the mechanisms of action of GPI that might lead to this hypothetical benefit are not clear, and biases may be responsible for this observation.

The improved TIMI grade flow and the improvement of TIMI grade flow ≥ 1 independent of P2Y12 type all argue for rapid and effective platelet inhibition by GPI. This underscores the importance of parenteral medications with reliable platelet aggregation inhibition in this particular patient population.

Another possible mechanism of GPI besides improving perfusion is inhibition of the direct interaction of platelets and leukocytes with the reperfused endothelium and reduced distal embolization of platelet aggregates, which have beneficial effects on the coronary microvasculature [13]. The potential anti-inflammatory effect of GPI may also be important. Inflammation plays a key role in MI, and several studies suggest that blockade of the GP IIb/IIIa receptor also limits the inflammatory response resulting from myocardial infarction and PCI [29]. C-reactive protein is known to increase significantly less in patients with MI in the first 24 h after abciximab and eptifibatide [29]. Unfortunately, we only had data on initial and maximum C-reactive protein levels during hospitalization, and there was no difference between the groups (*p* = 0.65) (unpublished data).

Bleeding has always been the major concern over the adjunctive use of GPI. The data on bleeding in patients with MI without CS are inconclusive. Both higher [3] and similar risks of bleeding have been observed [5]. The data from the era of potent P2Y12 are very limited for patients with CS [7,9,13]. Only 84 (55%) of our patients received P2Y12, which could affect clinical outcomes in these extremely high-risk patients. More than half of the patients who received P2Y12 received ticagrelor or prasugrel. The remaining 69 patients did not receive P2Y12 at all, probably because of their clinical condition, inability to swallow the tablets, or bleeding concerns. GPI was administered in 33 (47%) patients who did not receive P2Y12. P2Y12 could be administered via a nasogastric tube, usually crushed or dissolved, in mechanically ventilated patients [16]. Nasogastric tubes are usually not inserted in these patients on admission or during PCI but rather with some time delay after stabilization and/or transfer to the intensive care unit. Therefore, it is likely that in most patients with CS who are ventilated on admission, oral antiplatelet agents are fully activated at a time when GPI (when eptifibatide with its short half-life is used) already have a weaker antiplatelet effect, which may explain the similar bleeding rate in patients with or without P2Y12 in our analysis. The potentially deleterious effect of the combination of GPI and potent P2Y12 on bleeding may therefore have been less pronounced. As in previous studies in patients with CS, we did not find a higher bleeding rate in patients receiving GPI [8,9,13], nor did we find any difference in the bleeding rate in patients receiving P2Y12 (clopidogrel, prasugrel, or ticagrelor) or even potent P2Y12 with or without GPI. It is noteworthy that bleeding was associated with mortality. However, 90% of patients receiving GPI received eptifibatide and not abciximab. Eptifibatide is a competitive inhibitor of the GP IIb/IIIa receptor, and platelet aggregation normalizes approximately four hours after drug discontinuation, whereas abciximab is a Fab fragment that binds permanently to the GP IIb/IIIa receptor and inhibits platelet aggregation [30]. The use of eptifibatide in combination with the time delay of P2Y12 administration may further explain our finding. Very few patients received bivalirudin and there was no difference between the groups (Table 1). In addition, we performed a regression analysis which found no association between bivalirudin and bleeding risk.

In contrast to previous observations [8], patients receiving potent P2Y12 and GPI did not experience more bleeding than patients receiving potent P2Y12 without GPI. It must be remembered that the administration of GPI and P2Y12 in our analysis was at the discretion of the operator, the emergency physician, the physician on admission, and later the physician in the intensive care unit, hence it was not random. Patients at lower risk of bleeding may be preferentially selected for GPI and potent P2Y12. Indeed, patients receiving potent P2Y12 were less likely to have been resuscitated (59.6% vs. 74.8%), were less likely to have received oral anticoagulant therapy (7.8% vs. 25.0%), and were more likely to have had better GFR (63 mL/min/1.73 m^2^ vs. 52 mL/min/1.73 m^2^) (unpublished data). In addition, bleeding occurred in only 30.8% of patients receiving potent P2Y12 compared to 46.4% of all patients receiving all P2Y12 (clopidogrel, prasugrel, and ticagrelor), regardless of GPI. This supports the assumption that potent P2Y12 were carefully and selectively administered.

In contrast to previous findings, we did not observe an independent association between GPI and the risk of dying in one year [7,8,9,13]. We hypothesize that this was due to the use of a higher percentage of potent P2Y12 compared to previous analyses. P2Y12 were the only independent predictor of one-year mortality in our model.

Only 3.2% of our patients were treated using radial access. This is significantly less than in previous studies with CS [8]. However, all of our patients were mechanically ventilated, 70% after cardiopulmonary resuscitation, and 30% underwent therapeutic hypothermia. In such patients, there is a possibility that radial access will fail, necessitating conversion to femoral access (failure of puncture or failure of advancement with radial access) [31,32]. Even in prospective studies without the sickest patients, the conversion rate is 3.7%–7.6%, depending on the experience of the operator [32,33]. Crossover increases the procedure time, which could affect outcomes in high-risk patients. Significantly, radial access does not provide access for possible mechanical hemodynamic support (21% in our analysis), which may become extremely important in such patients [31,32]. All these factors make the low percentage of radial access more understandable. Although femoral access was predominantly used, the adjunctive use of GPI did not result in a higher bleeding rate, and our result cautiously supports the relative safety of GPI administration in these patients.

One of the problems of the analysis was that the population was gathered over a long period of time (11 years), and many differences in treatment may contribute to the changes in mortality over time (newer stents, potent P2Y12, bleeding, culprit lesion only PCI/multiple vessels PCI, mechanical circulatory support, etc.). In our analysis, multivessel PCIs were mainly performed at the beginning of the observation period and ended in approximately 5% in 2018 after the Culprit Shock study was published [34]. Current practice was not fully reflected in this study, which is certainly a limitation of the study. However, when we included year of admission in the multivariable analysis, it was not associated with either bleeding (*p* = 0.16 and *p* = 0.20 with potent P2Y12), or with TIMI 0/1 after PCI (*p* = 0.73, and *p* = 0.74 with potent P2Y12), so it is highly unlikely that these different practices over the years had a significant impact on the association between GPI and improved TIMI flow. On the contrary, only GPI were associated with TIMI 0/1 after PCI in this model (*p* = 0.035 and *p* = 0.036 with potent P2Y12). Despite all differences, we found in the multivariate analysis that GPI were associated with improved TIMI flow after PCI without increasing bleeding.

However, the results do not support the routine use of GPI in these patients. Rather, they provide data suggesting that, after individual consideration of bleeding risk or thrombotic burden, the additional use of GPI, even in combination with potent P2Y12, may have a beneficial effect without causing additional harm. An alternative antiplatelet agent for these patients is cangrelor, an intravenous reversible P2Y12 with a very short half-life and rapid onset and offset effects. Platelet function recovers completely approximately 60 min after discontinuation of the infusion [35]. Whether this additional intravenously administered antiplatelet agent should be a GPI or cangrelor needs to be clarified in future studies. However, it is difficult to imagine that there will ever be a randomized trial with these patients, and we will have to use the data from daily practice for the treatment of individual patients.

## 5. Conclusions

Due to the study design (limited sample size, retrospective inclusion with high risk of selection bias), our analysis does not allow us to draw conclusions about the effectiveness of GPI in this context. Despite all these limitations, GPI were associated with improved TIMI flow after PCI in our multivariable model without increasing bleeding rates. In addition, better CPC scores were observed, but no association between GPI and outcome was found. Our analysis suggests that selective use of GPI may be beneficial in mechanically ventilated patients with MI in CS without additional bleeding risk, even in the era of potent P2Y12.

## 6. Limitations

This was a retrospective observational study at a single tertiary center with all the associated limitations. The major limitation is the small sample size, especially for subgroup analyzes. In addition, the choice of the primary end point for this small sample size was ambitious because a large sample size is usually required to detect a significant difference in mortality and the number of patients may have affected the outcome. The use of GPI was at the discretion of the operators at the time of PCI and was not random, hence selection bias is possible. In addition, 90% of the GPI used were eptifibatide. However, abciximab is no longer available, so the information on eptifibatide may be relevant to daily practice in these patients. The decision to use GPI may have been influenced by several known or unknown variables that were not included in the analysis. Therefore, the study may not be powerful enough to detect all significant associations. Current practice was not fully considered in this study, which is an important limitation. We only had data on whether patients received P2Y12 but not the exact data on whether this occurred before/during/after PCI, and we lacked data on the possible administration of crushed/dissolved P2Y12 via nasogastric tube. In the vast majority of cases, femoral access was used. However, we would have expected even better results in terms of bleeding with radial access. We only followed patients for acute stent thrombosis. Our data included all-cause mortality only. Data on evidence-based medical therapy after PCI were not available. In addition, data on the drugs and doses used for anesthesia at the time of bleeding were not available. In multivariable analysis, wide CIs reduce the power of our analysis. There were no exclusion criteria related to concomitant diseases or clinical presentation, so this population represents a real experience of very high-risk patients requiring PCI.

## Figures and Tables

**Figure 1 jcm-11-07426-f001:**
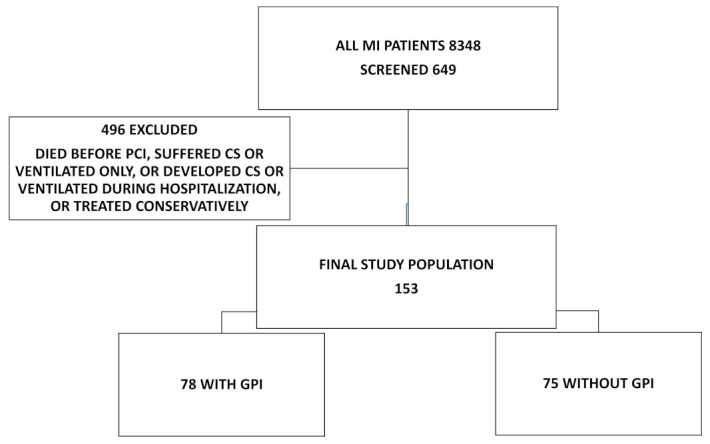
Study population. GPI = GP IIb/IIIa receptor inhibitor; MI = myocardial infarction.

**Figure 2 jcm-11-07426-f002:**
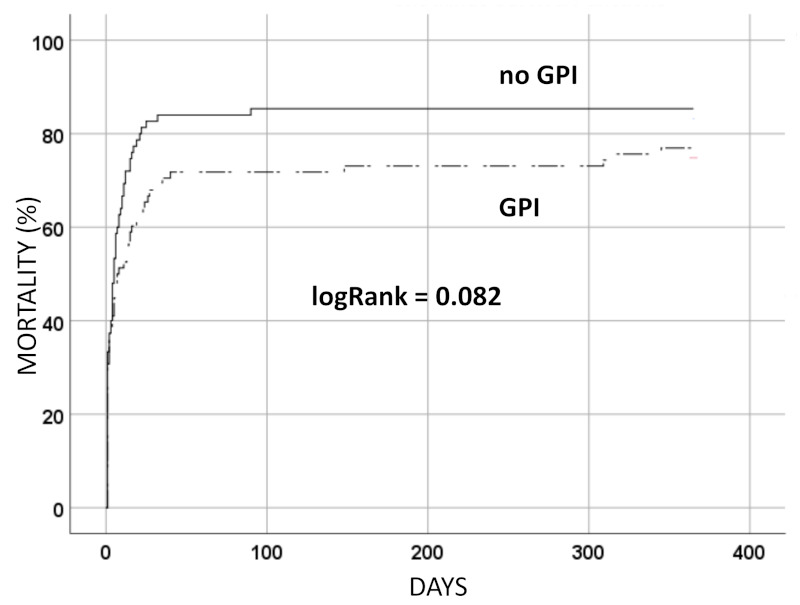
Unadjusted all-cause mortality at one year. GPI = glycoprotein IIb/IIIa receptor inhibitors.

**Table 1 jcm-11-07426-t001:** Baseline and procedural characteristics of the study population.

Variable	NO-GPI N = 75	GPI N = 78	POPULATION N = 153	*p*
Age (years)	66.0 (12.9)	65.1 (13.3)	66.0 (12.9)	0.66
Women	28 (37.5)	25 (32.1)	53 (34.6)	0.50
Diabetes	14 (18.7)	17 (21.8)	31 (20.3)	0.69
Hypertension	24 (32.0)	29 (37.2)	53 (34.6)	0.61
Hyperlipidemia	7 (9.3)	13 (16.7)	20 (13.1)	0.23
Smoking	12 (16.0)	11 (14.1)	23 (15.0)	0.82
Previous MI	13 (17.3)	10 (12.8)	23 (15.0)	0.50
Previous stroke	7 (9.3)	2 (2.6)	9 (5.9)	0.094
Previous PCI or CABG	7 (9.3)	2 (2.6)	9 (5.99)	0.094
Aortic stenosis	3 (4.0)	1 (1.3)	4 (2.6)	0.36
Known chronic kidney disease	7 (9.3)	4 (5.1)	11 (7.2)	0.36
STEMI	63 (84.0)	73 (92.3)	136 (88.2)	0.13
Anterior infarct	34 (45.3)	46 (59.0)	80 (52.3)	0.11
Cardiopulmonary resuscitation	50 (66.7)	58 (74.4)	108 (70.6)	0.37
Therapeutic hypothermia	19 (25.3)	37 (47.4)	56 (36.6)	0.007
Pulmonary edema	63 (84.0)	57 (73.1)	120 (78.4)	0.12
Hemoglobin (g/L)	128.2 (19.9)	130.8 (21.8)	129.6 (19.9)	0.055
Systolic pressure (mmHg)	85.6 (25.8)	91.5 (23.3)	88.6 (24.7)	0.14
Diastolic pressure (mmHg)	59.4 (16.2)	62.5 (14.2)	61.1 15.2)	0.23
Mean pressure (mmHg)	70.6 (17.0)	73.2 (15.7)	72.0 (16.3)	0.36
GFR (mL/min/1.73 m^2^)	49.4 (30.4, 64.1)	57.5 (44.4, 71.2)	53.0 (35.2, 68.9)	0.031
Radial access	3 (4.0)	2 (2.6)	5 (3.3)	0.68
PCI LMCA	13 (17.3)	15 (19.2)	28 (18.3)	0.84
PCI LAD	28 (37.3)	49 (62.8)	77 (50.3)	0.002
PCI LCX	11 (14.7)	22 (28.2)	33 (21.6)	0.05
PCI RCA	12 (16.0)	18 (23.1)	30 (19.6)	0.31
Multivessel PCI	15 (21.1)	26 (39.4)	41 (29.9)	0.025
Mechanical circulatory support	16 (21.3)	19 (24.3)	35 (20.9)	0.89
P2Y12	39 (52.0)	45 (57.7)	84 (54.9)	0.52
Potent P2Y12 (prasugrel, ticagrelor)	24 (32.0)	26 (33.3)	50 (32.7)	0.86
Bivalirudin	3 (4.0)	2 (2.6)	5 (3.3)	0.68
TIMI 0/1 before PCI	51 (68.0)	60 (76.9)	111 (72.5)	0.28
TIMI 0/1 after PCI	26 (34.7)	8 (10.3)	34 (22.2)	<0.0001
Tn, µg/L	53.6 (57.2)	63.1 (60.3)	58.9 (58.7)	0.44
EF	31.5 (5.9)	31.0 (4.5)	31.2 (5.2)	0.59
Bleeding	31 (41.3)	33 (42.3)	64 (41.8)	1.00
Bleeding in P2Y12 patients	18 (46.2)	21 (46.7)	39 (46.4)	1.00
Bleeding in potent P2Y12 patients	9 (37.5)	8 (30.8)	17 (34.0)	0.76
Acute stent thrombosis	3 (4.0)	1 (1.3)	4 (2.6)	0.36
CABG in the same hospitalization	6 (8.0)	2 (2.6)	8 (5.2)	0.16
CPC	2.86 (1.34)	1.57 (0.98)	1.89 (1.19)	0.011
Mortality outcome				
Death 30-day	62 (82.7)	54 (69.2)	116 (75.8)	0.06
Death one-year	64 (85.3)	60 (76.9)	124 (81.0)	0.22

Data are expressed as mean ± SD or as number (percentage) or as median (interquartile range). CABG = coronary artery bypass graft; CPC = cerebral performance category score; EF = ejection fraction; GFR = glomerular filtration rate; GPI = glycoprotein IIb/IIIa receptor inhibitor; LAD = left anterior descending coronary artery; LCX = circumflex artery; LMCA = left main coronary artery; MI = myocardial infarction; P2Y12 = P2Y12 receptor antagonist, PCI = percutaneous coronary intervention; RCA = right coronary artery; STEMI = ST-elevation MI; TIMI = thrombolysis in myocardial infarction; Tn = troponin.

**Table 2 jcm-11-07426-t002:** Predictors of 30-day and one-year mortality.

	30-Day Mortality		One-Year Mortality	
	OR (95% CI)	*p*	HR (95% CI)	*p*
Age	1.04 (0.99 to 1.08)	0.087	1.01 (0.99 to 1.03)	0.30
Male sex	1.55 (0.53 to 4.50)	0.42	0.87 (0.58 to 1.32)	0.52
Diabetes	0.82 (0.22 to 3.08)	0.77	0.96 (0.78 to 1.57)	0.87
Hypertension	0.59 (0.18 to 1.93)	0.38	1.03 (0.67 to 1.60)	0.88
Hyperlipidemia	2.30 (0.48 to 11.34)	0.29	0.99 (0.56 to 1.76)	0.98
GFR	1.00 (0.98 to 1.02)	0.99	1.00 (0.99 to 1.01)	0.99
STEMI	0.65 (0.10 to 4.28)	0.66	1.04 (0.58 to 1.86)	0.90
CPR	1.89 (0.50 to 7.24)	0.35	1.18 (0.72 to 1.92)	0.51
Therapeutic hypothermia	0.54 (0.14 to 2.11)	0.38	0.74 (0.45 to 1.22)	0.23
TIMI 0/1 after PCI	1.75 (0.42 to 7.25)	0.44	1.11 (0.68 to 1.80)	0.67
P2Y12	0.08 (0.02 to 0.32)	<0.0001	0.66 (0.45 to 0.98)	0.035
GPI	0.53 (0.19 to 1.46)	0.22	0.92 (0.62 to 1.34)	0.70
Bleeding	3.25 (1.17 to 9.05)	0.024	1.15 (0.77 to 1.72)	0.48

CI = confidence interval; CPR = cardiopulmonary resuscitation; DAPT = dual antiplatelet therapy; GFR = glomerular filtration rate; GPI = GP IIb/IIIa receptors inhibitors; HR = hazard ration; OR = odd ratio; P2Y12 = P2Y12 receptor antagonists; STEMI = ST-elevation myocardial infarction; TIMI = thrombolysis in myocardial infarction.

**Table 3 jcm-11-07426-t003:** TIMI grade flow before and after PCI.

	All Patients N = 153			All Patients Receiving P2Y12 N = 84			Patients Receiving Potent P2Y12 N = 50		
	No-GPI	GPI	*p*	No-GPI	GPI	*p*	No-GPI	GPI	
TIMI flow before PCI	0.84 (1.15)	0.62 (0.87)	0.17	0.82 (1.05)	0.56 (0.84)	0.20	0.71 (1.04)	0.42 (0.81)	0.28
TIMI flow after PCI	1.85 (1.27)	2.54 (0.85)	<0.0001	2.0 (1.26)	2.64 (0.77)	0.005	1.88 (1.33)	2.73 (0.60)	0.005
Improvement of TIMI flow ≥ 1	38 (50.7)	68 (87.2)	<0.0001	23 (59.0)	41 (91.1)	0.001	13 (54.0)	25 (96.2)	0.001

Data are expressed as mean (SD) or as number (percentage). GPI = glycoprotein IIb/IIIa receptor inhibitors; PCI = percutaneous coronary intervention; P2Y12 = P2Y12 receptor antagonists; TIMI = thrombolysis in myocardial infarction.

## Data Availability

The datasets used and/or analyzed during the current study are available from the corresponding author on reasonable request.

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
