# Peer review of "GP IIb/IIIa Receptor Inhibitors in Mechanically Ventilated Patients with Cardiogenic Shock due to Myocardial Infarction in the Era of Potent P2Y12 Receptor Antagonists"

_jcm, 2022, doi:10.3390/jcm11247426_

Round 1

Reviewer 1 Report

I would like to congratulate the authors for this interesting study dealing with a dramatic clinical situation and whose management deserves to be enlightened by relevant studies like this one. However, I would have a few comments to submit:

- the small sample size and retrospective nature of the study are major limitations to the interpretation of these results. Both of these points should appear at the beginning of the limitation section. 

- the choice of the primary endpoint is ambitious for this small sample size. This should also be emphasized.

- It seems important to me to emphasize the result observed on the improvement of the post-ICP TIMI flow. In my opinion, this is the main interpretable result of this study. In this sense, it is important to underline the prognostic character of this TIMI flow as described in the literature with for example (Mehta RH, Ou FS, Peterson ED, Shaw RE, Hillegass WB Jr, Rumsfeld JS, Roe MT; American College of Cardiology-National Cardiovascular Database Registry Investigators. Clinical significance of post-procedural TIMI flow in patients with cardiogenic shock undergoing primary percutaneous coronary intervention. JACC Cardiovasc Interv. 2009 Jan;2(1):56-64. doi: 10.1016/j.jcin.2008.10.006. PMID: 19463399.) 

- Could the authors highlight the results in terms of stent thrombosis?

- Is it possible to compare infarct size (TTE, biology...)?

- A few words on cangrelor and its place in this context could be written in the discussion 

-Given that the authors did not assess the CPC score in resuscitated patients undergoing therapeutic hypothermia with respect to GPI, I propose to temper and de-emphasize the better CPC in patients treated with IGP. The mechanisms of action of GPIs that may lead to this hypothetical benefit are clearly not elucidated. Biases may be responsible for this observation 

-Line 405 : please remove « We now have data on GPI in these patients » 

With regards

Reviewer 2 Report

The author Investigated the association between GP IIb/IIIa receptor inhibitors (GPI) and mortality and bleeding in patients with cardiogenic shock (CS) due to myocardial infarction (MI) who were mechanically ventilated on admission. There were several major limitations.

1. This study population was gathered over a long-term period. And, the current practice was not fully reflected in this study. one example, except for multivessel PCI, at least 50 patients in each group had PCI for the coronary artery, and in this situation, DAPT was essential for the antiplatelet therapy but only about 50% of the patient was prescribed P2Y12 inhibitor. These results may affect clinical outcomes in patients with high risk such as CS. Thus, the author should address this issue.

2. There are several evidence of GP IIb/IIIA receptor inhibitors regarding abciximab but not for the eptifibatide. The authors present this issue in the limitation section. But, the authors should address the difference between abciximab and eptifibatide. because most of the patients in the present study used eptifibatide.

3. Mortality is quite high but, it looks like most of the death cases occurred in 1 month. The author should add the no at risk in the KM curve.

Reviewer 3 Report

Dear Sirs,

I read with great interest the manuscript entitled:

„GP IIb/IIIa receptor inhibitors in mechanically ventilated patients with cardiogenic shock due to myocardial infarction in 3 the era of potent P2Y12 receptor antagonists”

I received for review an interesting, well-written manuscript on the care of patients with myocardial infarction complicated by cardiogenic shock. The still high mortality rate in this group of patients prompts the search for methods to optimise therapy. Such an attempt was made by the authors of this manuscript. Admittedly, the analysis is retrospective and the group included is small, but given the difficulties in setting up randomised trials on this group of patients, it adds to our knowledge in this area.

Sincerely,

Round 2

Reviewer 2 Report

The authors revised the manuscript in response to the comments but did not provide sufficient answers to the issues raised. Even considering it is a retrospective analysis, the reported conclusion couldn't agree due to insufficient rationale.
